# P-CSEM: An Attention Module for Improved Laparoscopic Surgical Tool Detection

**DOI:** 10.3390/s23167257

**Published:** 2023-08-18

**Authors:** Herag Arabian, Tamer Abdulbaki Alshirbaji, Nour Aldeen Jalal, Sabine Krueger-Ziolek, Knut Moeller

**Affiliations:** 1Institute of Technical Medicine (ITeM), Furtwangen University, 78054 Villingen-Schwenningen, Germany; 2Innovation Center Computer Assisted Surgery (ICCAS), University of Leipzig, 04103 Leipzig, Germany; 3Department of Mechanical Engineering, University of Canterbury, Christchurch 8041, New Zealand; 4Department of Microsystems Engineering, University of Freiburg, 79110 Freiburg, Germany

**Keywords:** attention module, laparoscopic video analysis, surgical tool classification

## Abstract

Minimal invasive surgery, more specifically laparoscopic surgery, is an active topic in the field of research. The collaboration between surgeons and new technologies aims to improve operation procedures as well as to ensure the safety of patients. An integral part of operating rooms modernization is the real-time communication between the surgeon and the data gathered using the numerous devices during surgery. A fundamental tool that can aid surgeons during laparoscopic surgery is the recognition of the different phases during an operation. Current research has shown a correlation between the surgical tools utilized and the present phase of surgery. To this end, a robust surgical tool classifier is desired for optimal performance. In this paper, a deep learning framework embedded with a custom attention module, the P-CSEM, has been proposed to refine the spatial features for surgical tool classification in laparoscopic surgery videos. This approach utilizes convolutional neural networks (CNNs) integrated with P-CSEM attention modules at different levels of the architecture for improved feature refinement. The model was trained and tested on the popular, publicly available Cholec80 database. Results showed that the attention integrated model achieved a mean average precision of 93.14%, and visualizations revealed the ability of the model to adhere more towards features of tool relevance. The proposed approach displays the benefits of integrating attention modules into surgical tool classification models for a more robust and precise detection.

## 1. Introduction

Over the years, advancements in technology have led to improved surgical procedures and, overall, to a better quality of patient care [1,2]. The introduction of these new technologies has, on the one hand, improved surgical efficiency as well as patient safety in operating rooms (ORs). Consequently, on the other hand, this has led to the ORs increased complexity [2,3]. This situation has paved the way for extensive research topics that try to harness the benefits of artificial intelligence (AI) to aid surgeons during operations [3]. Future ORs look to adopt the Industry 4.0 outlook, which utilizes smart systems i.e., AI solutions, and focus on providing real-time communication between surgeons and the multiple devices adopted during an operation for a smoother and more efficient workflow. A robust model that analyzes and interprets data collected from different sources is essential in providing an appropriate feedback for interventions, also referred to as computer-aided intervention (CAI).

CAI systems have the potential of improving surgical quality by providing a real-time feedback to the surgeons during an operation [2]. Surgical phase recognition is one of the applications that can help assist medical staff through skills assessment and protocol extraction, as well as to improve the management of ORs, e.g., preparing the next patient for surgery when observing the final phase of the current operation. Recognizing the different phases can be challenging; however, a correlation between surgical tool usage and phases has been observed [4]. Therefore, a robust surgical tool recognition model is necessary to provide a precise and accurate detection of the tools from the data captured using a laparoscopic camera. However, surgical tool classification models are not limited to simply phase recognition, but can also be utilized for different applications, i.e., camera zoom control [5].

Image-based data analysis is one of the most challenging topics in the field of machine learning. Current computer vision applications have adopted deep learning techniques, more specifically convolutional neural networks (CNNs), as research has shown they outperform traditional machine learning methods for object classification. In video-based image classification tasks, further improvements to the classification models have been introduced by considering temporal information from image sequences by means of recurrent neural networks (RNNs). In recent years, the concept of attention modules has also shown an increased presence in the literature due to their ability to improve model performance by refining spatial features with slight influence on computational complexity [6]. These attention modules can be integrated into any existing network architecture in order to improve the network representation and region of focus by concentrating on informative features and diminishing less important ones [6,7].

In this paper, an attention module, termed P-CSEM, was developed and evaluated to improve feature refinement and classification performance. First, the residual network architecture of ResNet50 [8] was selected as the base framework with initial weights trained on the ImageNet [9] database. ResNet50 was chosen following the results generated by the authors of [10], where a balance between the training time and performance was observed. Second, P-CSEM modules were added to the base ResNet-50. The proposed P-CSEM attention module was then evaluated against the attention modules of squeeze and excitation (SE) [6] and convolutional block attention module (CBAM) [7]. All models were trained and tested on the popular, publicly available Cholec80 database [4].

The paper is structured as follows: In Section 2, the methods used, attention module, network architecture, and analysis criteria are described. Section 3 highlights the key results while the discussions are rendered in Section 4. In Section 5, an ablation study conducted on the proposed model is reported, and a conclusion is drawn in Section 6.

### Related Work

The challenges of classifying surgical tools in real time have yielded different strategies, methods, and techniques, with more studies leaning towards deep learning approaches in recent times. In the study published by the authors of [4], surgical tool classification was tackled in a multi-task manner in combination with surgical phase recognition. Their study revealed an 81.0% mean average precision (mAP) on the Cholec80 dataset. To reduce the impact of the imbalanced distribution of surgical tools on training CNN models, a very deep CNN architecture model was implemented in the study published by the authors of [11], with data augmentation techniques achieving a mAP of 93.75%. In the study published by the authors of [12], a weakly supervised CNN pipeline was proposed with a novel tool localization maps approach. The proposed model achieved a mAP of 87.4%. A weakly supervised deep learning approach was proposed by the authors of [13] to perform surgical tool classification and localization by adding a multi-map localization layer and incorporating features at multiple stages within the architecture, which achieved a mAP of 90.6%.

In recent studies, modeling temporal information, along with surgical video sequences, showed great improvements over spatial approaches [14]. For instance, a spatial temporal network for surgical tool detection studied by the authors of [14] achieved state-of-the-art performance results, reaching a 94.74% mAP on the Cholec80 dataset.

In the study published by the authors of [5], a novel zoom control system was introduced. This system uses laparoscopic tool segmentation to extract tool geometry and perform automatic camera zoom adjustments based on the detection and position of specific tools for a better visual during surgery. In the study published by the authors of [15], a novel CNN architecture that generates ghost feature maps for a more efficient tool localization model by tackling the issue of redundant feature maps was studied. Performance results on the Cholec80 showed a 91.6% mAP and achieved an inference recognition speed of 38.5 frames per second (FPS).

Attention blocks have the ability to guide the decision-making process in the direction of a more refined feature space for a better classification performance in CNNs. Attention modules have been integrated in different network architectures and tasks ranging from image segmentation to classification. Their ability to retain informative features and suppress redundant ones makes them a critical part of supervised neural network training.

In the study published by the authors of [16], an attention-based temporal model was introduced for surgical phase recognition trained in an end-to-end approach. Results showed that this method was able to achieve an 89.8% accuracy with a 23.3 FPS inference time. A transformer-based model with attention, ‘OperA’, was proposed by the authors of [17] for surgical phase recognition. The performance of this model on the Cholec80 dataset was able to achieve an accuracy of 91.26%. In the study published by the authors of [18], an attention-guided network was examined for surgical tool detection on the m2cai-16 tool dataset, achieving a state-of-the-art mAP of 86.9%. In the study published by the authors of [19], a CNN model with attention was studied and evaluated on three different datasets. Performance results on the Cholec80 database achieved a mAP of 91.65% with a state-of-the-art, real-time recognition rate of 55.5 FPS.

In the study published by the authors of [20], a study on the performance of attention modules for laparoscopic tool classification was conducted. The results obtained showed a slight performance improvement when using attention modules over the base model. A visual inspection performed on a subset of images also showed the benefits of using attention modules in generating more focused informative features, thereby improving tool localization. In the study published by the authors of [21], a tool localization study was performed using an attention integrated network. Their results revealed that the attention-guided network had an increase in localization accuracy of around 30%, which stemmed from more focused class-aware spatial dimensions.

Following the achievements of attention-based techniques in improving base model performance, with slight changes in computation performance, attention modules were selected for this study’s surgical tool classification modeling approach, with the aim of achieving a robust and efficient classification model.

## 2. Materials and Methods

Two established attention modules, namely the squeeze and excitation (SE) [6] and convolutional block attention module (CBAM) [7], were evaluated for classifying surgical tools in laparoscopic images. Additionally, a new attention module, termed P-CSEM, was developed based on the advantages of the SE and CBAM modules to perform surgical tool classification.

The work performed by the authors of [6] proposed a SE block that investigates the inter-channel relations to improve network performance. Tests conducted on the ImageNet 2012 [9] dataset with different neural network architectures showed that incorporating the attention block into a given model outperformed the baseline results with small computational hindrance. An extension of the work performed by the authors of [6] by the authors of [7] yielded a new CBAM block that focused on spatial and channel interdependencies, highlighting that the spatial attention is ‘where’ the features of relevance are located within the feature map. Tests conducted on the ImageNet 1K [22] dataset showed the improved performance, and finer feature space, of the CBAM over both the base- and SE-infused models.

### 2.1. Squeeze and Excitation (SE) Block

One of the simplest, yet effective, attention modules is the squeeze and excitation (SE) module [6]. They have shown the potential to improve the representation ability of any network architecture through their ability to refine the channel features with minimal computational hindrance.

The features that enter the SE blocks are first squeezed to extract the channel hierarchy by means of a global average pooling (GAP). Two fully connected (FC) layers follow, with the first FC creating a bottleneck for the data by reducing the dimensionality by a factor of *R*. The reduction parameter R was selected to be 16 in this study, as smaller *R* values showed reduced model performance (Section 5.2). The second FC expands the data back to the original dimension, which is then passed through a sigmoid activation layer to bound the feature data between 0 and 1. The output is then multiplied with the input features to the SE block, thereby producing a more refined feature space with a concentration on the more informative feature maps. Figure 1 shows the SE attention block architecture.

### 2.2. Convolutional Block Attention Module (CBAM)

Convolutional block attention modules (CBAMs) [7] are an extension of the SE block and have shown improvements in classification tasks. The CBAM not only considers the channel significance, but also the spatial feature relevance in deciding ‘where’ to concentrate.

The CBAM channel-wise attention is similar to that of SE with the addition of a global max pooling (GMP), that runs parallel to the GAP, to infer more relevant features. The features then pass through two FCs reducing and rescaling the dimension as in the SE block structure. The refined features are then multiplied with the input features, which is later passed through the spatial attention module that performs a spatial average pooling (SAP) and a spatial max pooling (SMP) across the channels. The output then passes through a convolutional layer with a 7 × 7 filter size and a number of filters set to one. Following this step, a sigmoid activation function is performed before multiplying the output with the input feature space, thereby achieving a refined feature space of both spatial and channel focus. Figure 2 presents the CBAM attention block architecture.

### 2.3. Parallel-Convolutional Squeezed and Excitation Module (P-CSEM)

The parallel-convolutional squeezed and excitation module (P-CSEM) is a new attention module that has been introduced in this work. The P-CSEM is composed of two series blocks that run in parallel to each other. The first series block incorporates the advantages of the SE block by refining the feature space across the channel domain, thereby highlighting the strongest map of influence, from the multitude of channels. This structure is coupled with the second series block, which makes use of the CBAM concept to extract the area of focus from the feature space by applying a SAP operation. Following this, the features from the SAP are excited to a higher dimensional space via a convolutional layer of filter size 3 × 3 and number of channels 8, equivalent to the number of tools (7 tools) plus one representing no tool. The output then passes through a rectified linear unit (ReLU) activation function before being entered into another convolutional layer, which squeezes the channel dimension back to one with a filter size of 3 × 3. The outputs from both series blocks are then multiplied together.

The input features are then rescaled with the new feature space from the series blocks to achieve better refinement before moving on in the network architecture. The structure of the P-CSEM is depicted in Figure 3, along with the activation dimensions at each block. The reduction parameter (*R*) used to reduce the feature space was set to 16 following the results of an ablation study (as detailed in Section 5.2).

### 2.4. System Modeling

To highlight the ability of attention modules to improve network outcomes, a comparison between four models, one base, and three attention infused models was conducted. The CNN architecture of ResNet50 [8] with pre-trained weights, that was trained on the ImageNet [9] dataset, was selected as a base model. ResNet50 was chosen based on the results published by the authors of [10,20,23] that showed the efficacy of residual networks in classifying surgical tools.

The base model of ResNet50 is composed of five convolutional blocks, with varying residual blocks in each, followed by a GAP and an FC layer. The attention modules are flexible, such that they can be placed in different positions within any existing architecture. In this work, the attention modules of SE, CBAM, and P-CSEM were integrated into the base model after the second, third, and fourth convolutional blocks, respectively, after addition operator i.e., layer and prior to the ReLU activation. The ResNet50 is composed of 177 layers with a 224 × 224 × 3 image input and a total of 25.6 million parameters. The network architecture, along with the placement of the attentions, is depicted in Figure 4.

The attention modules were placed in the described position to achieve the best possible data refinement while balancing the computational and model performance. The features that are generated in the early layers, i.e., in convolution block 1, of the architecture do not significantly contribute towards the class-specific direction, but more of a general feature extraction, and therefore it was not considered as a location to include attention. In the following 2nd, 3rd, and 4th convolution blocks, the features became more class specific and, therefore, a guidance towards more relevant and informative features should be considered. The top layers of the network deal with a feature space that is small in spatial dimension and class specific; therefore, an attention module at this stage would be contrary to the performance improvement.

An ablation study was performed to validate the placement methodology chosen. The addition of attention modules within the convolution blocks, i.e., in the residual blocks, was considered. This resulted in the addition of 16 attention modules throughout the network architecture. The use of four attention modules during the early stages of the architecture, i.e., in the residual blocks of the 2nd and 3rd convolution, was also examined as per the study published by the authors of [20].

### 2.5. Training Settings

The models were run under a MATLAB 2021a environment (The MathWorks, Natick, MA, USA), on a desktop with an Intel Xeon @ 2.20 GHz (Intel^®^, Santa Clara, CA, USA), 64.00 GB memory (RAM), and 64-bit Windows 10 operating system (Microsoft Corporation, Redmond, WA, USA) with an NVIDIA graphics card GeForce RTX 2080Ti (NVIDIA Corporation, Santa Clara, CA, USA). A varying learning rate, beginning with 0.002 and decaying at a rate of 0.0009 per iteration, was run for 10 epochs for model training. A stochastic gradient decent with momentum (SGDM) optimization function was used with a batch size of 50 images and cross-entropy loss function.

### 2.6. Evaluation Criteria

The presence of multiple tools in one image transforms the inference into a multi-label classification task. In order to evaluate the model’s ability in recognizing more than one class in an image, the sigmoid activation function was used prior to the classification layer. The sigmoid output corresponds to a probability between 0 and 1 for each class. The model was trained on the first 40 videos and tested using the last 40 videos of the Cholec80 dataset.

To evaluate the performance of the models, the mean of the average precision (mAP) of each of the tools was determined across the testing set. A network explain-ability analysis based on a strategy adopted by the authors of [21], using gradient-weighted class activation mapping (Grad-CAM) [24], was performed separately. Equation (1) describes the computation of the evaluation metric for AP:(1)$$\begin{array}{c} AP_{c}=\int_{a}^{b}f\left(x_{c}\right)dx={\left(\frac{1}{2}\overset{N-1}{\underset{n=0}{\sum }}\left(f\left(x_{n+1}\right)+f\left(x_{n}\right)\right)\Delta_{x_{n}}\right)}_{c} \end{array}$$
where c denotes the class number. $AP_{c}$ is the average precision of the model for a certain class c; the recall boundary values are set as a = 0 and b = 1. $x_{c}$ represents the recall of class *c*, and $f\left(x_{c}\right)$ is the corresponding precision value of c. N is the total number of points of $x_{c}$. The recall and precision of point 𝑁 is calculated based on the different probability thresholds determined using an exponential curve of values between 0 and 1.

## 3. Results

### 3.1. Database Description and Distribution

The publicly available Cholec80 database was chosen for the training and testing of the different models. The database consists of 80 cholecystectomy procedures that were recorded at the University Hospital of Strasbourg. The video recordings were captured at a rate of 25 Hz with tool annotations every 1 FPS. Seven surgical tools (grasper, hook, bipolar, scissors, clipper, irrigator, and specimen bag) were present in the database. The tool annotation conditions were such that at least half of the tool tip should be observed in the labeled frame [4]. The database is constituted of 184,498 image frames from the 80 videos. Figure 5 represents an illustration of the different surgical tool tips of (a) Grasper, (b) bipolar, (c) hook, (d) scissor, (e) clipper, (f) irrigator, and (g) specimen bag, available in the Cholec80 dataset [4].

Table 1 shows a summary of the distribution of the images from the Cholec80 dataset into the different classes and sets. The surgical tools, grasper, and hook displayed dominance over both the training and testing sets, with a combined presence of more than 83% of the entire data.

To evaluate the performance of the tool tip localization prediction of the models, bounding boxes were annotated for the videos numbered from 41 to 45. These annotations represent the ground truth for the localization assessment.

The EndoVis 2019 [25] challenge dataset was used for the generalizability evaluation. This dataset included classes that were not present in the Cholec80 database, and therefore these classes and, consequently, the image frames, were removed from processing.

### 3.2. Model Performance

As can be seen from the AP and mAP results from Figure 6, the P-CSEM attention module outperformed the other models in each tool category except for the irrigator class. A significant improvement can be observed between the base and P-CSEM model for the grasper tool, with an increase in AP of around 7.5%, as well as the scissors tool with around 4%. The P-CSEM-integrated model achieved a mAP of 93.14% compared to 91.38% and 91.57% of the SE and CBAM models, respectively, also achieving a 1.76% improvement over the base model.

A closer look into the activations of each of the four models was taken. The models were able to separate the different components from the image, with the P-CSEM and SE-integrated models achieving a more comprehensive and refined selection.

The inference times for the classification of a single image frame was calculated for each model. The experiments were carried out with GPU and CPU. The P-CSEM model achieved a recognition rate of 158 Hz and 29 Hz, the base model 166 Hz and 42 Hz, the SE model 165 Hz and 34 Hz, and the CBAM model 155 Hz and 30 Hz, with GPU and CPU, respectively.

## 4. Discussion

As shown in Figure 6, the P-CSEM attention module was able to achieve a better performance over the base model in nearly all tools. The increase in the scissor tool was noted to be due to the underrepresentation of this class in the class distribution; therefore, the attention module was able to highlight the particular features of this tool in order to boost its recognition rate. The significant increase in the Grasper tool was attributed to a more refined feature space thanks, in part, to the attention module. The learning process guided by the attention module was able to distinguish between the tips of a grasper and a clipper as they are quite similar in structure. The reduced performance seen in the irrigator was linked to the shape and design of the tool, as in most images, the tip of the irrigator was often hidden under tissue, thereby making it difficult to be recognized. It is important to mention the strength in recognizing the correct tool, which was attributed to the fine refined feature space that is more precise to tool specifications rather than the general shape of the instrument.

Following the activations and Grad-CAM evaluation, the P-CSEM-integrated model was able to separate the different components better than the base model. The more the features were refined, as in the P-CSEM model, the more focused the true-positive prediction becomes, due to a more separable decision boundary, reducing the likelihood of focusing on areas that are not specific to the tool tip. The P-CSEM also displayed robustness to noise, such as light reflection.

Table 2 shows the mAP performance results of different methods on the Cholec80 dataset. As observed, the ResNet50 + P-CSEM model performed well, achieving the runner-up position in the list of non-temporal tool recognition methods. It is important to mention that the best-performing model was trained on a substantially large number of iterations (70 K) compared to the 17,270 iterations used to train the proposed ResNet50 + P-CSEM model. The state-of-the-art (SOTA) also used a different training-to-validation ratio, coupled with data augmentation and data elimination, compared to the implemented approach of the P-CSEM. The network architecture used in the SOTA was also very deep, at 164 layers, which increases the possibility of overfitting the model given the high number of iterations and considering the use of transfer learning for initial weights.

A re-implementation of the SOTA architecture using the training and testing conditions set in this study was conducted. The results revealed that the re-implemented SOTA achieved an accuracy of 89.17%, which is less than that of the base model results reported in this work.

The P-CSEM and base models were evaluated against a different dataset of cholecystectomy tools from the EndoVis 2019 [25] challenge. This evaluation was performed in order to observe the generalizability of the trained model on data from a different source. The experiments revealed that the P-CSEM model achieved a mAP of 59.52%, while that of the base model achieved a mAP of 58.57%. Although these models were not able to achieve comparable mAP results as in the Chole80 dataset, their results were deemed acceptable considering the differences in tool shape and design.

The attention-integrated model described in this study performed well on the Cholec80 dataset. Certain limitations were considered for this study, such as the fixed dataset used for testing, imbalanced class distribution, and model explain-ability. To tackle these issues, future work will involve using a larger dataset and the incorporation of the temporal information. To increase the size of the dataset further, and address the imbalanced class distribution, image augmentation techniques could be applied. The inference rates also revealed that any one of these models can be integrated and used in a real-time environment, as image capture rates in laparoscopic surgeries typically range between 50/60 Hz.

## 5. Ablation Study

### 5.1. Attention Integration

The placement of the attention block within the existing architecture is important to the overall performance improvement of the model. To that end, different locations and combinations were tested for their optimal performance. First, in Place 1, the attention blocks were placed after each residual block within the ResNet50 architecture. This adds more to the computational power by having 16 attention blocks added to the entire framework. The second placement group, Place 2, was conducted after the 2nd, 3rd, and 4th convolutional blocks, as previously described in Section 2.4. Here, three attention blocks were incorporated into the framework. The last placement, Place 3, was set inside the convolution blocks of two and three, following the first and second residual blocks of each. This placement added four attention blocks to the existing architecture.

Table 3 represents the attention integration results. The mAP reveals that Place 2 achieved the best performance in both the P-CSEM and SE; although, the margin between the results was small, with most instances having a less than 1% difference. The computational burden was also considered as a factor, and with placement 2 having the least number of blocks added to the existing architecture, making the computational load less than that of the other placements, while still achieving comparable results.

### 5.2. Reduction Parameter

The reduction parameter (R) is also a critical selection for how much information is retained. The selection of the parameter for this study followed the work by Hu et al. [6], which was 16. An ablation study was carried out on the P-CSEM to highlight the impact of this factor on the recognition process. Three reduction rates of four, eight, and sixteen were selected for R. The reduction parameter analysis was only carried out on the P-CSEM, since it showed the best overall performance, as observed in the Results Section 3.2.

Table 4 represents the results of the ablation study performed on the reduction parameter (R) for the P-CSEM module. As observed, the reduction rates of four and eight had a higher computational training time as well as parameter count. Although the mAP results revealed that the reduction parameter of eight performed better than with sixteen, the improvement was seen as negligible with a difference of around 0.07% for the mAP. When comparing the parameter count and the duration of training, the reduction parameter of 16 performed optimally with a faster training time and least number of parameters.

## 6. Conclusions

In this study, a new attention module was introduced that captures and refines relevant spatial and channel features, working in parallel to improve the performance of surgical tool recognition. The attention-infused model of P-CSEM was able to outperform the base and other attention modules by a 1.76% margin on the Cholec80 dataset. The proposed method was able to achieve second place among the state-of-the-art; however, it utilized a fraction of the training time and iteration count of the first place model. The reduced time and resources make this attention module desirable in improving the performance of base network architectures.

## Figures and Tables

**Figure 1 sensors-23-07257-f001:**
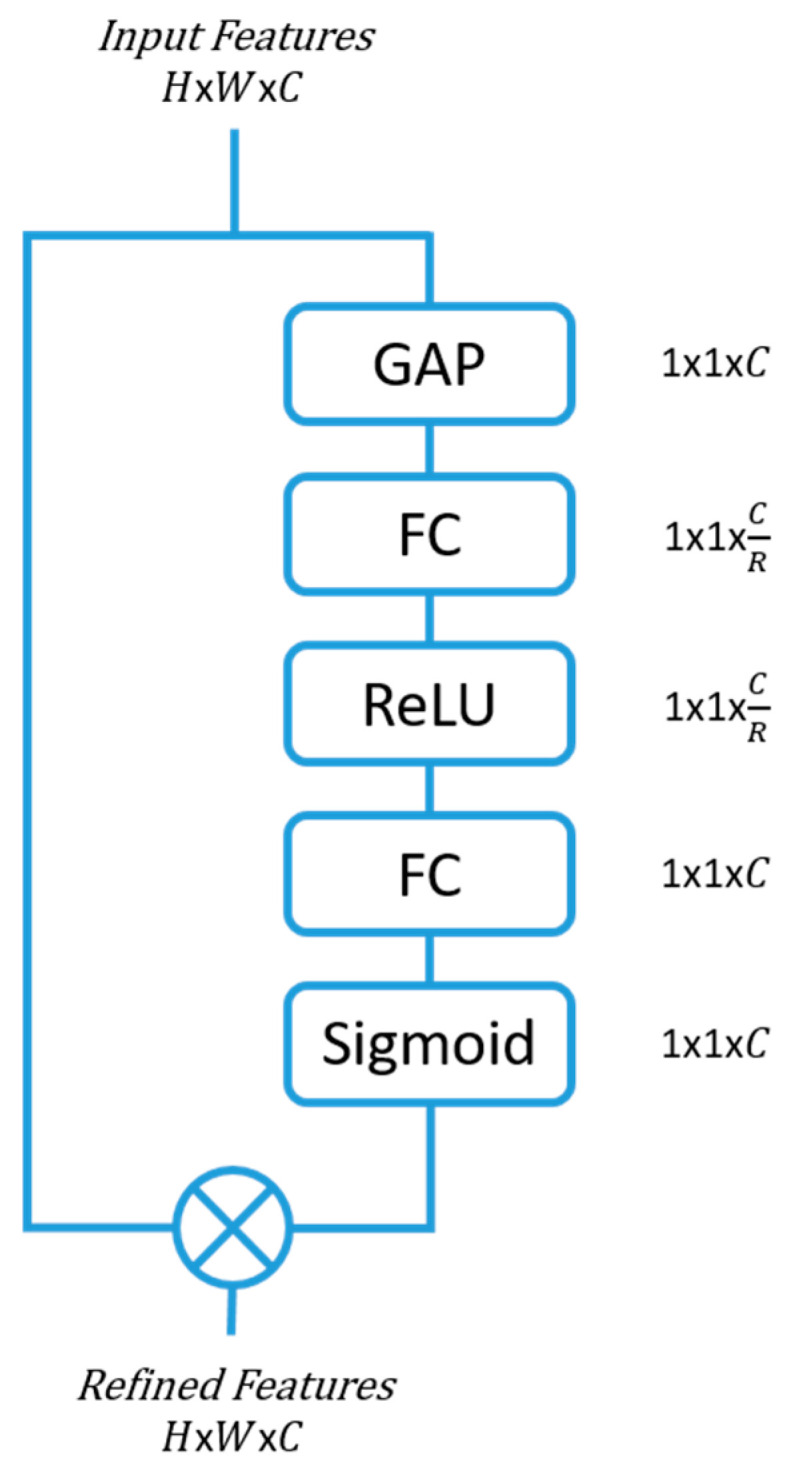
Squeeze and excitation (SE) attention module architecture. Global average pooling (GAP), fully connected (FC), and rectified linear unit (ReLU). *C* is the channel, *H* is the height, *W* is the width of the dimensional space, and *R* is the reduction parameter.

**Figure 2 sensors-23-07257-f002:**
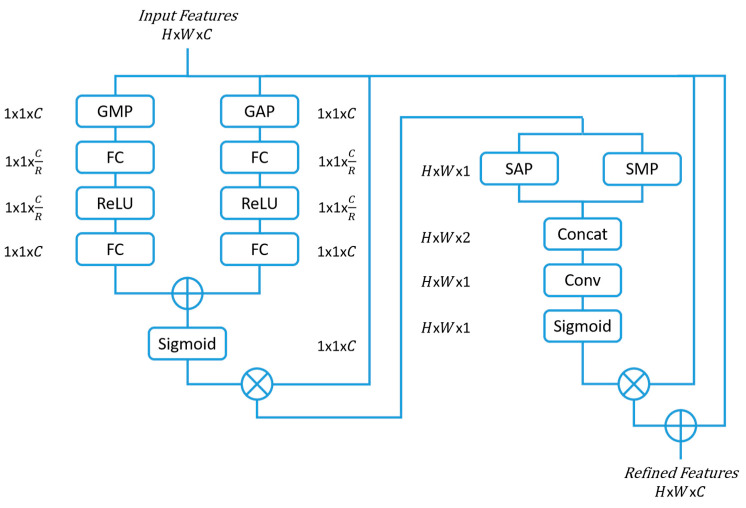
Convolutional block attention module (CBAM) architecture. Global average pooling (GAP), global max pooling (GMP), fully connected (FC), rectified linear unit (ReLU), spatial average pooling (SAP), spatial max pooling (SMP), channel-wise concatenation (Concat), and convolution with 7 × 7 filter (Conv). *C* is the channel, *H* is the height, *W* is the width of the dimensional space, and *R* is the reduction parameter.

**Figure 3 sensors-23-07257-f003:**
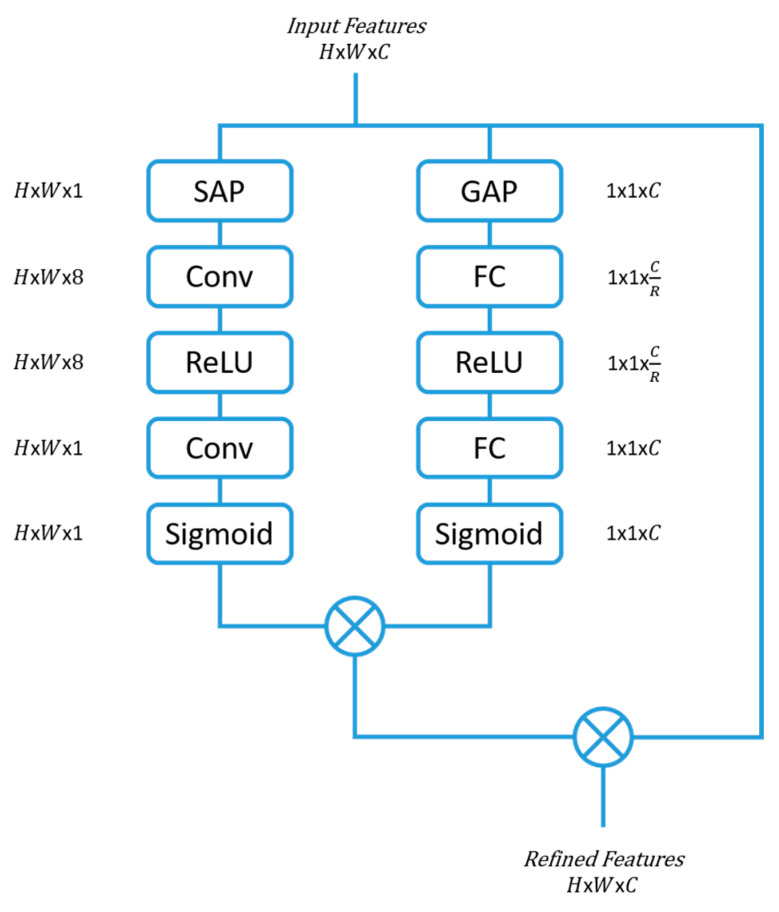
P-CSEM attention module architecture. Global average pooling (GAP), fully connected (FC), rectified linear unit (ReLU), spatial average pooling (SAP), and convolution with 3 × 3 filter (Conv). *C* is the channel, *H* is the height, *W* is the width of the dimensional space, and *R* is the reduction parameter.

**Figure 4 sensors-23-07257-f004:**
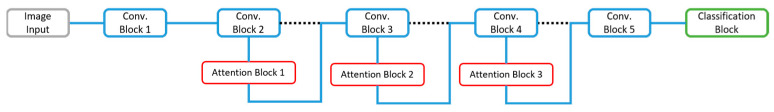
ResNet50 architecture with the attention modules placements. Black dotted lines indicate where the connection was cut from the base model, while the blue solid lines indicate the flow of the network.

**Figure 5 sensors-23-07257-f005:**
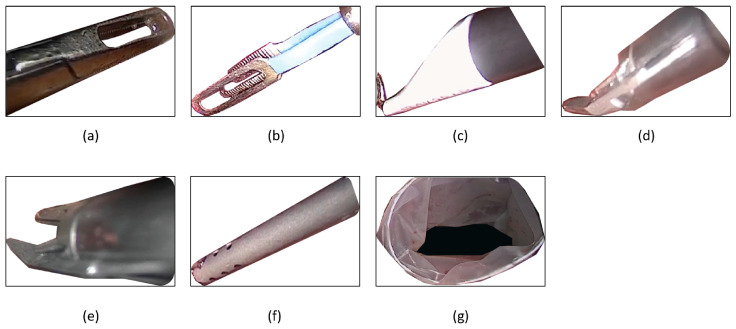
Samples images of the surgical tool tips present in the Cholec80 database. (**a**) Grasper, (**b**) bipolar, (**c**) hook, (**d**) scissor, (**e**) clipper, (**f**) irrigator, and (**g**) specimen bag.

**Figure 6 sensors-23-07257-f006:**
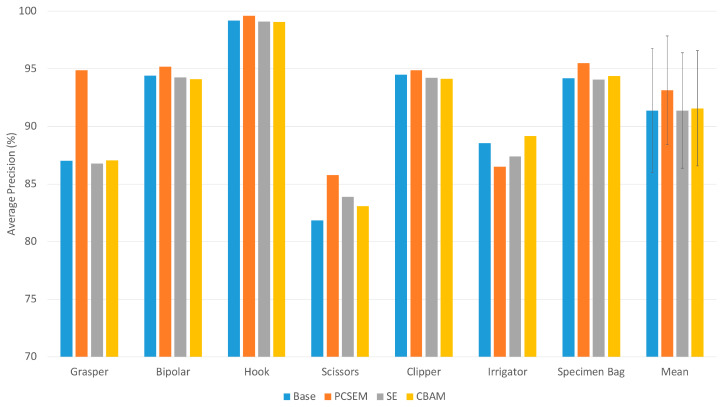
Average precision results of each tool and mean average precision (mAP) of the testing set for each model.

**Table 1 sensors-23-07257-t001:** Distribution of images from the Cholec80 database into their respective classes.

	Training	Testing
Grasper	56,800	45,788
Bipolar	4106	4770
Hook	48,437	54,669
Scissors	1624	1630
Clipper	3217	2769
Irrigator	5384	4430
Specimen bag	5760	5702
Total	86,304	98,194

**Table 2 sensors-23-07257-t002:** Comparison of the mAP ± standard deviation results from different methods on the Cholec80 dataset. Value in bold represents the best performance. Values in () represent the results from the method re-implementation.

	mAP (%)
Twinanda et al. [4]	81.00 ± 11.84
Jaafari et al. [11]—original	**93.75** ± 5.84 (89.17 ± 8.83)
Shi et al. [19]	91.65
Vardazaryan et al. [12]	87.40 ± 17.21
Yang et al. [15]	91.60
ResNet50 {base model}	91.38 ± 5.41
ResNet50 + SE	91.38 ± 5.02
ResNet50 + CBAM	91.57 ± 5.00
ResNet50 + P-CSEM	93.14 ± 4.72

**Table 3 sensors-23-07257-t003:** mAP (%) results of attention block integration of the different modules. Values in bold represent the best performances.

	# of Attention Blocks	ResNet50 + P-CSEM	ResNet50 + SE	ResNet50 + CBAM
Place 1	16	91.50	90.94	91.03
Place 2	3	**93.14**	**91.38**	91.57
Place 3	4	91.01	91.10	**91.73**

**Table 4 sensors-23-07257-t004:** mAP results, duration of each model’s training, and number of parameters for each reduction parameter for the ResNet50 + P-CSEM model. Values in bold represent the best performances.

Reduction Parameter (R)	mAP (%)	Time (h:m)	Parameters
4	93.18	12:47	24.2 M
8	**93.21**	12:50	23.9 M
16	93.14	**12:10**	**23.7 M**

## Data Availability

The database used in this study was (Cholec80). The Cholec80 dataset is available (http://camma.u-strasbg.fr/datasets/ (accessed on 22 March 2017)) from the respected publisher upon request.

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
