# Peer review of "P-CSEM: An Attention Module for Improved Laparoscopic Surgical Tool Detection"

_sensors, 2023, doi:10.3390/s23167257_

Round 1

Reviewer 1 Report

The authors submitted an interesting manuscript. The manuscript is very well written. It uses an excellent English. Further, the manuscript is well illustrated with high-quality figures. It was clearly described what was done. Figure and table captions are informative and self-contained. The structure of the paper is clear and logical. It is easy to read and follow the manuscript.

I think the manuscript can be accepted for publication after a minor revision.

i) In the database description, it would be good to illustrate the applied database with several images. Probably, not everybody is familiar with surgical tools.

ii) Publication of training curves would be nice, since deep learning involves a lot of experiments.

iii) Can the proposed method be used in a real environment? Is it robust against image noise?

Reviewer 2 Report

In this work, an attention module, termed P-CSEM, was developed and evaluated to improve feature refinement and classification performance OR. It is an interesting study and all important parameters have been evaluated and correlated to the obtained data. This paper can be published after a minor revision.

1.     In fig. 5, standard error bars are missing. Please add it.

2.     The proposed method was able to achieve the second place among the state of the arts, however, it utilized a fraction of the training time and iteration count of the first place model. The required explanations for this improvement is not given in the manuscript.

3.     Is there any way the quality of image or pixels will affect the data quality in this method. What is the relationship between figure quality and testing data. Pl. explain it.

4.     In Table 2, standard deviation values are not given, so there are only few % changes between the best and poor data. I believe that may be in the standard deviation of error range. Pl. explain it

5.     There are some typo errors.

6.     Some references are incomplete, for example ref 2, 28, please and provide complete details.
